# Barriers and facilitators of adherence to long-term antiretroviral treatment in Kampala, Uganda

Stephen Okoboi [1]*, Andrew Mujugira[1,2], Nicolate Nekesa[1], Barbara Castelnuovo[1], Sheri A. Lippman[3], Rachel King[1,4]

1 Research Department, Infectious Diseases Institute Makerere University, Kampala, Uganda,
2 Department of Global Health, University of Washington, Seattle, Washington, United States of America,
3 Department of Medicine, Division of Prevention Science, University of California, San Francisco, California, United States of America, 4 Department of Global Health, University of California San Francisco, San Francisco, Calufornia, United States of America

* okoboi25@gmail.com

## Abstract

Few qualitative studies have evaluated adherence to long-term antiretroviral treatment (ART) in sub-Saharan Africa. We explored adherence barriers and facilitators among PLWH on long-term ART (≥10 years in Kampala)., We conducted 22 in-depth interviews with 16 purposively selected PLWHs on long-term ART and six purposively selected healthcare providers at the Infectious Diseases Institute ART clinic.,. Interviews with PLWH explored their perspectives regarding; comprehension of ART adherence, ART adherence experiences, and adherence barriers and facilitators. Provider interviews covered: perceived ART adherence barriers and facilitators and how to reinforce and support ART adherence. Qualitative data were analyzed using a deductive content analytic approach. The median age of PLWH was 52 years [IQR] 39 - 65). Most (63%) had been on ART for 15-20 years, 50% were male, and 38% had treatment supporters. Both providers and PLWH on long-term ART emphasized the importance of adhering to prescribed medication to suppress HIV. Adherence facilitators: (i) Achieving sustained viral suppression improved overall health and motivated PLWH to maintain long-term adherence. (ii) Spousal treatment partners, financial support from adult children for transportation to clinic appointments, and the desire to fulfill social responsibilities were also adherence motivators. (iii) Policy changes, such as providing multi-month drug refills and community ART delivery, were effective strategies for promoting adherence. Adherence barriers: (i) Financial constraints – lack of money for transportation – often resulting in delayed or missed appointments to the clinic. (ii) Work obligations that conflicted with clinic appointments negatively impacted long-term ART adherence. (iii) Pill fatigue and avoidance of adherence counseling sessions by PLWH with detectable viral load exacerbated non-adherence. Familial support and differentiated ART delivery had a mutually beneficial effect on long-term adherence. Future studies should evaluate the unique adherence needs of this rapidly growing ART-experienced population and identify the most efficient strategies for adherence support.

**Data availability statement:** I have removed all potentially identifiers particularly all the three variables of health workers (age, educational level and gender). This can easily enable health workers to be identifies since they are the only staff dedicated to cohort study.

**Funding:** SO was supported by the Fogarty International Center of the National Institutes of Health (NIH) under Award Number D43TW009343 and the University of California Global Health Institute (UCGHI). The content is solely the authors' responsibility and does not necessarily represent the NIH's or UCGHI's official views. BC was partly supported by the Fogarty International Centre, National Institute of Health (grant# 2D43TW009771-06 "HIV and coinfections in Uganda"). The funders had no role in study design, data collection and analysis, publication decisions, or manuscript preparation.

**Competing interests:** The authors have declared that no competing interests exist.

## Introduction

In 2022, Uganda had an estimated 1.4 million people living with HIV (PLWH) (range, 1.3–1.6 million [1]. Of these individuals, over 1.3 million were receiving antiretroviral therapy (ART) in 2022 [2]. ART has transformed HIV from a life-threatening illness to a chronic condition, leading to improvements in both longevity and quality of life [3–5].

Antiretroviral medications not only help PLWH to maintain good health and a lifespan comparable to those without HIV, but they also offer the possibility for PLWH with undetectable viral load to engage in condomless sex with minimal risk of transmitting HIV to their partner(s) [6]. Over two decades of research have consistently shown that ART is highly effective in reducing sexual transmission of HIV [7–9], providing robust evidence for the "Undetectable = Untransmittable" (U=U) concept endorsed by the Joint United Nations Programme on HIV/AIDS (UNAIDS) [10,11]. However, achieving and sustaining viral suppression requires lifelong adherence to effective ART and continuous monitoring. This can prove challenging for PLWH on long-term ART, who may experience treatment fatigue and comorbidities that impact their adherence behaviors [12–14] and make consistent long-term adherence a potential barrier to U=U.

The first PLWH globally to receive ART through the assistance of the U.S. President's Emergency Plan for AIDS Relief (PEPFAR) in 2003 was from Uganda [3,15]. The Infectious Diseases Institute (IDI), a large PEPFAR implementing partner, and The AIDS Support Organization were among the first facilities in the country to offer comprehensive HIV care, support, and treatment services that same year. The IDI ART clinic has followed a cohort of 1,000 PLWH since 2003/2004 [16,17]. PLWH in this cohort undergo regular medical assessments, receive adherence counseling, and engage in an adherence action plan review every three months [18]. Previous studies have amply documented the barriers and facilitators of ART adherence among PLWH. However, these studies mainly focused on PLWH, who had been on ART for less than ten years [13,19,20]. This study assessed the barriers and facilitators of long-term ART adherence (>10 years) at the IDI ART clinic in Kampala, Uganda.

## Methods

### Population and procedures

The IDI HIV Centre of Excellence, situated within the premises of Mulago Teaching Hospital, is a large outpatient clinic that serves over 8,000 PLWH in Kampala, Uganda. It hosts the ART Long-Term (ALT) cohort [19,21]. From 3rd January to 30th June 2022, we conducted in-depth interviews with 22 participants,16 purposively selected PLWHs on long-term ART, and six purposively selected healthcare workers (three nurses, one lay counselor, and two doctors) at the IDI ART clinic. Interviews with PLWH explored their perspectives regarding (i) comprehension of ART adherence, (ii) ART adherence experiences, and (iii) barriers and facilitators to adherence. Provider interviews covered (i) perceived ART adherence barriers and facilitators and (ii) how to reinforce and support ART adherence through facilitators such as treatment supporters.

### Sampling and data collection

Drawing from the literature indicates that saturation can be achieved with 9 -17 interviews [22]. We purposively sampled 22 participants. Of these, 16 were PLWH, and six were healthcare providers. We selected PLWH in the long-term ART cohort who met three criteria: 1) participants who attended scheduled appointments at the IDI adult HIV clinic, 2) 18 years or older, and 3) being on ART for a minimum of 10 years. The ALT cohort nurse purposively

selected the PLWH to be interviewed from the list of scheduled appointments. The study also purposively included healthcare workers who had provided ART services for at least three years but were primarily dedicated to managing ALT cohort participants. The clinic leadership introduced the female research assistant (RA) to providers and PLWH. She had a background in social sciences and experience in qualitative data collection. She approached the study participants and provided them with an explanation of the study. None declined to participate. Distinct interview guides were used for healthcare providers and PLWH, which included questions based on previous research on barriers and facilitators of ART adherence [19]. The RA conducted the interviews in the Mulago HIV clinical room. Each interview lasted 46-65 minutes and was audio recorded in English or Luganda, the widely spoken language in Central Uganda. The RA then translated the Luganda interviews into English. We interviewed all 16 PLWHs and six HCWs as planned, and this number of participants provided sufficient information to reach thematic saturation.

## Data analysis

We analyzed the data using a deductive content analytic approach centered on descriptive category development [21]. We used NVivo 11 (QSR International, Melbourne, Australia) for systematic data management. Data coding proceeded as follows: (i) all transcripts and field notes during interviews were independently read by the investigator (SO) and the RA (NN) to identify major codes. The independent analysis prevented selective perception and interpretive bias and ensured rigor of the coding process; (ii) a selection of transcripts was double-coded and subsequently compared. Any discrepancies were resolved, resulting in a codebook that included descriptive labels for each coded category, elaborative narratives, and key informant interview (KII) quotes illustrating the underlying concepts; and (iii) using the constant comparison method, we developed codes to capture categories that emerged in the data. The analysis continued iteratively, refining the major categories and codes via discussion between authors SO and NN to identify barriers and facilitators of long-term ART adherence. Study findings are reported using the Consolidated Criteria for Reporting Qualitative Research Findings [23].

## Ethical approval

This study was approved by the Infectious Diseases Institute Research Ethics Committee (IDI REC-041/2021) and the Uganda National Council for Science and Technology (HS1896ES). We formally obtained written informed consent from the participants.

## Results

### Participant characteristics

Table 1 shows that the median age of the 16 PLWH was 52 years (interquartile range [IQR] 39-81) for PLWH and 38 years (IQR 28-52) for the six providers. Eleven PLWH (69%) had been on ART for 16-20 years, and the median ART duration was 17 years (IQR 10-20). Nine PHWH (56%) had secondary-level education (> seven years of schooling), 12 (75%) were not formally employed, and eight (50%) were male. Ten (60%) had no documented detectable viral load (>1,000 copies/ml) according to Uganda National guidelines on viral load suppression cutoff while on ART, and 40% had active treatment supporters (S1 Data).

### Qualitative results

We present the results in two broad categories of *facilitators* and *barriers* to long-term ART adherence. Adherence facilitators: Achieving viral suppression motivated long-term

**Table 1. Participant Characteristics.**

| PLWH (N=16) | Median (IQR) or N (%) |
|---|---|
| Gender | |
| **Male** | 8 (50%) |
| **Females** | 8 (50%) |
| Age (years) | 52 (39-81) |
| Occupation | |
| **Formally employed** | 4 (25%) |
| **Not employed** | 12 (75%) |
| Education Level | |
| **Primary level** | 3 (19%) |
| **Secondary level** | 9 (56%) |
| **Tertiary level** | 4 (25%) |
| Active treatment supporters | |
| **Yes** | 6 (40%) |
| **No** | 10 (60%) |
| Duration on ART | |
| **10 - 15 years** | 5 (31%) |
| **16 - 20 years** | 11 (69%) |
| Detectable viral load | |
| **Yes** | 4 (40%) |
| **No** | 10 (60%) |
| Healthcare workers (N=6) | |
| Age (years) | 38 (28-52) |
| Gender | |
| **Male** | 4 (67%) |
| **Female** | 2 (33%) |
| Occupation | |
| **Nurse** | 3 (50%) |
| **Doctor** | 2 (33%) |
| **Counsellor** | 1 (17%) |

adherence and led to improved overall health. Fulfilling family responsibilities also encouraged ART adherence and, ultimately, extended survival. Having a supportive spouse was crucial in maintaining long-term adherence to ART. Additionally, the parent-child relationship and financial support from adult children were essential in sustaining long-term ART adherence. Adherence coping strategies suggested by providers empowered PLWH and reinforced positive adherence behaviors. Policy changes such as multi-month drug refills and community delivery were effective in boosting ART adherence. Adherence barriers: Financial hardship resulted in delayed or missed clinic appointments. Work obligations negatively impacted long-term ART adherence, as did skipping scheduled follow-up counseling sessions by PLWH with detectable viral load.

## Category 1: Facilitators of long-term ART adherence

*Suppressed viral load motivated long-term adherence and improved overall health.* The motivation for adhering to ART was driven by the understanding that following a daily regimen of antiretroviral medication would lead to viral suppression, which participants described as viral dormancy. This, in turn, contributed to better overall health and well-being, a gradual but recognizable

outcome. The visible improvement in appearance, "looking good," and seeing non-adherent peers decline and look "worn out and emaciated" also incentivized PLWH to maintain their treatment. The discovery that they were virologically suppressed further emphasized the critical role of good adherence in maintaining viral inactivity and preventing viral rebound.

> "*They tell me that the virus is dormant; it has also helped me to see that I take my drugs consistently to see that it doesn't become active again if God helps me such that it gets dormant completely*" [Female, age 48].

*The ability to fulfill social responsibilities encouraged ART adherence and living longer.* The ability to fulfill familial obligations was a significant motivating factor in maintaining long-term adherence to ART, leading to increased longevity. For widowed women, living longer was particularly important as they often served as crucial pillars of support for their families. Women living with HIV expressed concerns over the well-being of their families in the event of their passing or compromised health. They feared paternal relatives would not offer adequate care and support to their children.

> "*For that reason, I have to adhere well to my medication because there are people I brought into this world, my children, who are entirely dependent on me. So, part of my goal is that I have children who are entirely looking at me. I am their dad, their mother*" [Female, age 65].

> "*I have to work for my children because no one will care for them; their paternal relatives don't. Everyone cares for themselves*" [Female, age 54].

*Having a spouse as an adherence supporter was pivotal in sustaining long-term ART adherence.* Having a designated treatment buddy, such as a trusted family member or friend, played a vital role in promoting ART adherence among male PLWH on long-term ART. This effect was particularly evident when the "supporter" was a spouse who was also undergoing HIV treatment. Together, spouses reinforced each other's medication routines, promoted open communication about treatment information, and prioritized attending clinic appointments. This shared experience of adhering to ART extended to jointly managing medications and potentially facilitated disclosure of HIV status within the partnership.

> "*Then I take it home (drugs), and my wife and I handle it (the drugs) together because both of us are on drugs*" [Male, age 53].

> "*My treatment buddies ensure that I take my drugs. They can even do a pill count at home and find out whether I am taking medicines or not*" [Male, age 52].

*The parent-child relationship and parent's financial support were essential in sustaining long-term ART adherence.* Establishing healthy relationships between parents and children encouraged the disclosure of parental HIV status to their children. This fostered trust and facilitated open communication between child and parent, ultimately providing the parent with peace of mind. Additionally, adult children played a vital role in financially supporting their parents on treatment, alleviating the burden of transportation costs, and promoting adherence to ART by enabling them to attend refill appointments as scheduled.

> "*I stay far away, but my daughter provides transport to bring me here. I have never missed any of my appointments. I won't lie to you. This has also helped me greatly because it gives me peace of mind.*" [Female, age 56].

*Adherence coping strategies empowered PLWH and reinforced adherence behaviors.* Health-care professionals noted that personalized medication-coping strategies positively impacted the adherence behaviors of PLWH on long-term ART. These coping strategies, recommended by healthcare workers, empowered PLWH to take their medications consistently. As a result of this learned behavior, their overall health improved, and their adherence to ART was reinforced as they aged.

*"The way I understand adherence to ART is like empowering the patient and making the patient cope with their medication, coping with the drugs yourself and deciding to take the drugs the way you have been advised to take and accepting fully"* [Male Doctor age 52].

*Changing the frequency of drug refills to multi-month and community delivery boosted ART adherence.* Regular visits to the clinic for clinical monitoring, medication refills, and transportation difficulties posed a burden for PLWH on long-term ART. This was exacerbated by the lengthy wait times at the clinic for appointments and refills. However, modifying the frequency of medication refills to every 3-6 months in both community-based models and health facilities significantly improved access to treatment, reducing the number of required visits from 12 annually to only 2-4 per year. Additionally, this change addressed challenges related to transportation. The updated policy allowing for multi-month refills in both community-based and health facility models has greatly facilitated ART refills.

*"The fact that when you go to pick drugs, it's a one-time thing, probably once a month. I used to pick drugs monthly, then started coming every three months, but apparently, I am now coming after every six months. You cannot fail to get transport for once in six months"* [Female, age 48].

## Category 2: Barriers to long-term ART adherence

*Living in poverty led to delayed or missed clinic appointments.* Limited financial resources resulted in insufficient funds for transportation, leading to delayed or missed clinic appointments and medication refills. These financial limitations were further exacerbated by an inability to purchase necessary food, causing the individual living with HIV to forgo taking their medication even when it was accessible. This lack of proper nutrition further weakened their health, leading to the inability to manage their treatment effectively.

*"I have already explained that my emphasis is on financial status. It is up and down. Financially, if somebody is assured of financial assistance, that person could do an extremely good job. But because some finances are up and down, you find yourself in a situation where you don't have enough food. These drugs have their consequences. You can get dizzy, you can feel weak just because of bad feeding"* [Male, age 50].

*Work routines negatively affected long-term ART adherence.* The confluence of daily medication management and work schedules negatively affected ART adherence, notably when the designated time for taking medication coincided with work hours. When faced with the need to administer their medicines at the workplace, both female and male PLWHs resorted to discreetly taking their drugs, often avoiding detection by colleagues to prevent inadvertent disclosure and potential stigmatization related to HIV.

*"The nature of work can be a barrier to adherence in that one can fail to get time to swallow medication when at work. Their time for taking medication may reach when they are too*

*busy. Some people hide while taking HIV medication when he/she is with colleagues to whom he has never disclosed their status because they fear their colleagues seeing them, especially if they still have a stigma. So, it becomes difficult for them to take their medication because one has to be free and accept their status to take their medication well"* [Male, age 56].

*Pill fatigue was expressed as feeling worn out and exhausted.* During clinic check-ups and medication refill appointments, healthcare providers observed that PLWH on long-term ART often expressed feelings of being "worn out and exhausted" due to the lifelong daily regimen. This sense of pill fatigue was linked to a lack of motivation to adhere to their prescribed medications, which was compounded by the added responsibility of managing other conditions typically diagnosed by healthcare professionals.

*"Others are tired of taking treatment because they have been on treatment for more than ten years, others more than 17 years, others more than 18 years"* [Female nurse, age 32].

*Missing scheduled follow-up counseling sessions was perceived as non-adherence.* Healthcare providers observed that some PLWHs were referred for counseling but did not follow through with attending the session. This avoidance of counseling sessions was often rooted in an underlying belief that there was "something wrong with ART adherence." Consequently, this negative perception caused PLWHs to develop a dismissive attitude towards counselors and healthcare workers who recommended counseling. Furthermore, PLWHs on long-term ART may have perceived themselves as stable and, therefore, did not see any value in participating in counseling.

*"Some patients have a negative attitude towards the counselor. When told that "they are going to see a counselor," they may think they have done something wrong, so some shy away from seeing counselors. Some may leave without seeing a counselor"* [Female counselor, age 45].

## Discussion

This qualitative study explored barriers and facilitators of long-term ART adherence among PLWH at a large HIV clinic in Kampala, Uganda. The findings indicated that the facilitators and barriers to long-term ART adherence involve complex interactions between the individual, their family, and the healthcare system. Facilitators of adherence included sustained viral suppression, which not only motivated long-term adherence but also led to improved overall health. Additionally, the desire to fulfill family responsibilities encouraged ART adherence and ultimately extended survival. The presence of a supportive spouse was crucial for maintaining long-term ART adherence. Policy changes, such as multi-month drug refills and community delivery of ART, were also effective in promoting adherence. Barriers to adherence included financial constraints, specifically lack of money for transportation, resulting in delayed or missed clinic appointments. Work obligations that conflicted with clinic appointments negatively impacted long-term ART adherence, as did skipping follow-up counseling sessions by clients with detectable viral loads.

Our data suggest that individual factors such as having a suppressed viral load and using adherence coping strategies facilitated long-term ART adherence. A sub-Saharan Africa ART adherence systematic review (involving 161,922 PLWH from 146 articles) also found that those who experienced improved health status were likelier to adhere to their medication [24]. Furthermore, another systematic review and meta-analysis (207 articles and 103,836 PLWH) found that coping strategies and self-efficacy – defined as a person's belief in their capacity to perform a given behavior needed to reach a desired goal – were the most influential factors

associated with ART adherence [19]. The current ART program in Uganda faces challenges due to the increasing number of PLWH initiated on ART due to the test and treatment program compared with the earlier HIV program, which included extensive preparation and adherence counseling before starting treatment. The limited ART psychosocial preparedness in test and treatment programs could lead to increased non-adherence.

Maintaining long-term adherence requires continuous medication refills and a holistic care approach for PLWH on long-term ART. Our research revealed that strong parent-child relationships and financial support from children for transportation to clinic appointments were important factors in facilitating regular medical reviews and medication refills. However, as seen in other studies [25–27]. The monotonous routine of taking daily pills over an extended period can lead to weariness, resulting in pill fatigue among PLWH [25]. Moreover, the demands of work may make it difficult for PLWH on ART to find time to take their medication or may force them to conceal their medication use due to fear of unintended disclosure and HIV-related stigma [25]. Our study found that insufficient income to meet nutritional requirements and refill prescriptions posed an additional obstacle to long-term ART adherence. This is particularly relevant for PLWH, who often experience high levels of food insecurity [26,27]. Studies have shown that poverty and food insecurity are associated with poor ART adherence [26,27]. Research indicates that PLWH who miss their ART refill appointments are twice as likely to experience food insecurity compared to those who adhere to their medication schedule [26,27]. The introduction of injectable antiretroviral drugs, currently unavailable in sub-Saharan Africa, may help address some of these barriers at the individual level.

In Africa, the family plays a critical role as a microsystem for social support in promoting long-term ART adherence [28]. When family members are supportive, understanding, progressive, and cohesive, it creates an environment that encourages PLWH to adhere to medication and fulfill their family responsibilities, such as caring for children, the elderly, or dependents [29]. Our research aligns with previous studies highlighting the positive impact of having a spouse as an adherence supporter on long-term ART adherence [30]. Treatment supporters like spouses provide ongoing reminders for drug refills and ensure timely pill taking while also fostering open communication and motivation for long-term ART adherence [30]. Studies in Uganda and South Africa have shown that PLWH without treatment supporters were more likely to have detectable viral load from non-adherence to medications [31].

Our finding that health system factors like multi-month ART refill duration (3-6 months) facilitated long-term ART adherence is consistent with earlier qualitative research among both PLWH and healthcare providers in South Africa, which also found that 6-month medication refills positively influenced PLWH adherence due to the convenience of only having to access medications twice a year [32].

One strength of our study is its contribution as one of the initial qualitative evaluations of ART adherence among PLWH who have been on treatment for at least a decade. By conducting interviews with both female and male PLWHs, as well as providers, we were able to gather diverse perspectives on adherence. However, it is important to note that our findings may not be generalizable to other settings due to the nature of our study being conducted in an urban location at the IDI ART clinic, a prominent regional center for HIV care.

## Conclusions

Barriers and facilitators of long-term ART adherence were similar to those encountered during short-term treatment, highlighting the pervasive impact of these multi-level factors. PLWH on long-term ART should receive continuous adherence assessment, counseling, and support to address pill fatigue and HIV-related stigma. Future research should evaluate the

specific adherence needs of this rapidly expanding demographic of individuals on long-term ART and identify the most effective strategies for sustaining adherence and managing pill fatigue associated with the emergence of other chronic conditions.

## Supporting information

**S1 Data. Shows the demographic dataset of the people living with HIV.** (XLSX)

## Author contributions

**Conceptualization:** Stephen Okoboi, Andrew Mujugira, Rachel King.

**Formal analysis:** Stephen Okoboi, Nicolate Nekesa, Barbara Castelnuovo.

**Funding acquisition:** Stephen Okoboi, Barbara Castelnuovo, Rachel King.

**Methodology:** Stephen Okoboi, Andrew Mujugira, Rachel King.

**Project administration:** Stephen Okoboi.

**Supervision:** Stephen Okoboi, Barbara Castelnuovo, Rachel King.

**Writing – original draft:** Stephen Okoboi.

**Writing – review & editing:** Stephen Okoboi, Andrew Mujugira, Barbara Castelnuovo, Sheri A. Lippman, Rachel King.

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
