## [Decision Letter · Decision Letter 0]

9 Dec 2024

PGPH-D-24-02254

Barriers and Facilitators of Adherence to Long-Term Antiretroviral Treatment in Kampala, Uganda

Dear Dr. Okoboi,

Thank you for submitting your manuscript to PLOS Global Public Health. After careful consideration, we feel that it has merit but does not fully meet PLOS Global Public Health’s publication criteria as it currently stands. Therefore, we invite you to submit a revised version of the manuscript that addresses the points raised during the review process.

We look forward to receiving your revised manuscript.

Kind regards,

Janet Seeley

Academic Editor

Journal Requirements:

Additional Editor Comments (if provided):

Thank you very much for this interesting paper. Reviewer 2 has requested some additional information on the sampling as well as some rephrasing. I would be grateful if you could attend to those comments.

Reviewers' comments:

Reviewer's Responses to Questions

**Comments to the Author**

1. Does this manuscript meet PLOS Global Public Health’s publication criteria ? Is the manuscript technically sound, and do the data support the conclusions? The manuscript must describe methodologically and ethically rigorous research with conclusions that are appropriately drawn based on the data presented.

Reviewer #1: Yes

Reviewer #2: Yes

2. Has the statistical analysis been performed appropriately and rigorously?

Reviewer #1: N/A

Reviewer #2: N/A

3. Have the authors made all data underlying the findings in their manuscript fully available (please refer to the Data Availability Statement at the start of the manuscript PDF file)?

Reviewer #1: Yes

Reviewer #2: No

4. Is the manuscript presented in an intelligible fashion and written in standard English?

Reviewer #1: Yes

Reviewer #2: Yes

5. Review Comments to the Author

Reviewer #1: This is clearly written, and the researcher describes the methodology quite well.

The research focused on the experiences and perspectives of PLWH who had been on treatment for over a decade. This makes the study unique because most of the studies conducted earlier in different settings had participants who had been on ART for less than 10 years.

In addition, the study sought to understand the perspectives of the health providers concerning the barriers and facilitators that patients experienced with adherence on the long-term treatment.

The study used a qualitative approach for data collection and participants were selected purposively following a set criterion that included having been on ART for at least 10 years and attending the IDI clinic in Mulago and aged 18 years or older. This is a useful study.

Reviewer #2: This is a well designed qualitative study and the results are presented clearly.

I only have two recommendations

1) Explain why you decided to interview 16 PLWH and 6 HCPs. You mention in the paper that you reached data saturation but it is not clear why you decided on these numbers initially and whether you stopped interviewing based on saturation - that said saturation is increasingly being challenged in qualitative research as an arbitrary justification for qualitative sample size decisions. Therefore I would prefer to know why you decided to interview this number of participants in the first place.

2) You use the term "PLWH interviews" - it would be preferable to talk about 'interviews with PLWH"

Overall it is a good paper and worthy of publication.

6. PLOS authors have the option to publish the peer review history of their article (what does this mean? ). If published, this will include your full peer review and any attached files.

**Do you want your identity to be public for this peer review?** For information about this choice, including consent withdrawal, please see our Privacy Policy .

Reviewer #1: **Yes: ** Agnes Ssali

Reviewer #2: **Yes: ** Mitzy Gafos

---

## [Decision Letter · Decision Letter 1]

31 Jan 2025

Barriers and Facilitators of Adherence to Long-Term Antiretroviral Treatment in Kampala, Uganda

PGPH-D-24-02254R1

Dear Dr Okoboi,

We are pleased to inform you that your manuscript 'Barriers and Facilitators of Adherence to Long-Term Antiretroviral Treatment in Kampala, Uganda' has been provisionally accepted for publication in PLOS Global Public Health.

Best regards,

Janet Seeley

Academic Editor

Thank you very much for attending to the comments. I look forward to seeing this paper in its published form.

Reviewer Comments (if any, and for reference):

Reviewer's Responses to Questions

**Comments to the Author**

1. If the authors have adequately addressed your comments raised in a previous round of review and you feel that this manuscript is now acceptable for publication, you may indicate that here to bypass the “Comments to the Author” section, enter your conflict of interest statement in the “Confidential to Editor” section, and submit your "Accept" recommendation.

Reviewer #2: All comments have been addressed

2. Does this manuscript meet PLOS Global Public Health’s publication criteria ? Is the manuscript technically sound, and do the data support the conclusions? The manuscript must describe methodologically and ethically rigorous research with conclusions that are appropriately drawn based on the data presented.

Reviewer #2: Yes

3. Has the statistical analysis been performed appropriately and rigorously?

Reviewer #2: N/A

4. Have the authors made all data underlying the findings in their manuscript fully available (please refer to the Data Availability Statement at the start of the manuscript PDF file)?

Reviewer #2: No

5. Is the manuscript presented in an intelligible fashion and written in standard English?

Reviewer #2: Yes

6. Review Comments to the Author

Reviewer #2: (No Response)

7. PLOS authors have the option to publish the peer review history of their article (what does this mean? ). If published, this will include your full peer review and any attached files.

**Do you want your identity to be public for this peer review?** For information about this choice, including consent withdrawal, please see our Privacy Policy .

Reviewer #2: **Yes: ** Mitzy Gafos
